# QTL mapping of seedling and field resistance to stem rust in DAKIYE/Reichenbachii durum wheat population

Shitaye Homma Megerssa[1,2]*, Karim Ammar[3], Maricelis Acevedo[4], Gary Carlton Bergstrom[5], Susanne Dreisigacker[3], Mandeep Randhawa[6], Gina Brown-Guedira[7], Brian Ward[7], Mark Earl Sorrells[2]

1 Ethiopian Institute of Agricultural (EIAR), Addis Ababa, Ethiopia, 2 Plant Breeding and Genetics Section, School of Integrative Plant Science, Cornell University, Ithaca, NY, United States of America, 3 International Maize and Wheat Improvement Center (CIMMYT), Mexico D. F., Mexico, 4 Department of Global Development, Cornell University, Ithaca, NY, United States of America, 5 School of Integrative Plant Science, Plant Pathology and Plant-Microbe Biology Section, Cornell University, Ithaca, NY, United States of America, 6 International Maize and Wheat Improvement Center (CIMMYT), Nairobi, Kenya, 7 USDA-ARS Plant Science Unit, Raleigh, NC, United States of America

* shhomete@gmail.com

**Data Availability Statement:** The data has been deposited in Figshare and can be accessed at the following links: Field response of RILs to stem rust

## Abstract

Stem rust caused by the fungus *Puccinia graminis* f.sp. *tritici* Eriks. & E. Henn. (*Pgt*) threatens the global production of both durum wheat (*Triticum* t*urgidum* L. ssp. *durum* (Desf.) Husnot**)** and common wheat (*Triticum aestivum* L.). The objective of this study was to evaluate a durum wheat recombinant inbred line (RIL) population from a cross between a susceptible parent 'DAKIYE' and a resistant parent 'Reichenbachii' developed by the International Center for the Improvement of Maize and Wheat (CIMMYT) 1) for seedling response to races JRCQC and TTRTF and 2) for field response to a bulk of the current *Pgt* races prevalent in Ethiopia and Kenya and 3) to map loci associated with seedling and field resistances in this population. A total of 224 RILs along with their parents were evaluated at the seedling stage in the Ethiopian Institute for Agricultural Research greenhouse at Debre Zeit, Ethiopia and in the EIAR and KALRO fields in Ethiopia and Kenya, for two seasons from 2019 to 2020. The lines were genotyped using the genotyping-by-sequencing approach. A total of 843 single nucleotide polymorphism markers for 175 lines were used for quantitative trait locus (QTL) analyses. Composite interval mapping (CIM) identified three QTL on chromosomes 3B, 4B and 7B contributed by the resistant parent. The QTL on chromosome 3B was identified at all growth stages and it explained 11.8%, 6.5%, 6.4% and 15.3% of the phenotypic variation for responses to races JRCQC, TTRTF and in the field trials ETMS19 and KNMS19, respectively. The power to identify additional QTL in this population was limited by the number of high-quality markers, since several markers with segregation distortion were eliminated. A cytological study is needed to understand the presence of chromosomal rearrangements. Future evaluations of additional durum lines and RIL families identification of durable adult plant resistance sources is crucial for breeding stem rust resistance in durum wheat in the future.

and genotype data URL: https://figshare.com/
articles/dataset/QTL_mapping_of_seedling_and_
field_resistance_to_stem_rust_in_DAKIYE_
Reichenbachii_durum_wheat_population/
20326143 Seedling response of RILs to stem rust
and genotype data URL: https://figshare.com/
articles/dataset/QTL_mapping_of_seedling_and_
field_resistance_to_stem_rust_in_DAKIYE_
Reichenbachii_durum_wheat_population_
Seedling_phenotype_and_genotype_data/
20619036.

**Funding:** This study was supported by the DGGW
project funded by the UK Aid from the British
People and the Bill & Melinda Gates Foundation.
The award number was OPP1133199. The funders
have no role in the design of the study, data
collection, analysis, and have no publication
decision except that the paper must be published in
an open access journal.

**Competing interests:** The authors declared that no
competing interests exist.

## Introduction

Durum wheat (2n = 4x = 28, AABB genome) is a tetraploid wheat species used for the industrial processing of pasta and other food recipes, mainly consumed in Mediterranean and North African regions [1,2]. The processing of end-use products from durum wheat demands both grain yield and quality. However, several factors constrain these and other agronomically important traits. Stem rust *Puccinia graminis* f.sp. *tritici* Eriks. & E. Henn. (*Pgt*) is one of the most devastating diseases of both common wheat and durum wheat [3]. The stem rust fungus draws assimilates from the vascular system and results in reduced grain yield and shriveled seeds that reduce end-use product quality [4,5]. This pathogen can also cause a complete yield loss when susceptible varieties are grown under environmental conditions conducive for disease development [6]. The commonly used stem rust control methods are the use of genetic resistance and application of fungicide. In the presence of genetic variability for resistance, genetic resistance is the preferred method due to its advantage in environmental safety and cost efficiency [7].

Many of the commercially deployed stem rust resistance genes are qualitative or race specific. The extensive deployment of qualitative resistance is often challenged by continuously evolving virulent races causing resistance to be ineffective [8,9]. Races in the Ug99 group and other virulent races unrelated to Ug99 with broad virulence to several *Sr* genes in wheat cultivars threaten global wheat production and food security [10,11]. Ug99 (TTKSK) overcame the resistance conferred by *Sr31*, a resistance gene that had been effective over three decades. Race TKTTF broke the resistance confered by *SrTmp* and caused the 2013/14 epidemic in Ethiopia. This race devastated a popular bread wheat variety called 'Digalu' planted on more than 100,000 ha. Durum wheat lines carrying *Sr13* are reported to be resistant to races TTKSK and TKTTF [12,13]. However, races JRCQC idenitifed in Ethiopia in 2009 and TTRTF, first identified in Georgia in 2014, later causing the 2016 stem rust epidemic in Sicily, Italy carry combined virulence to *Sr13b* and *Sr9e*. These two alleles are widely deployed in durum wheat cultivars produced in different parts of the world including CIMMYT and North American durum wheat germplasm [14,15]. However, the *Sr13a* allele is effective against these races (JRCQC and TTRTF) [16,17]. Wide deployment of *Sr13a* in durum wheat may risk breakdown of resistance by emerging virulent races. Therefore, the identification of molecular markers linked to quantitative trait locus/loci (QTL) and incorporation of new sources of resistance is essential for the succesful resistance breeding strategies for wheat stem rust.

New variability can be introduced to breeding lines from wild relatives and/or landraces. Durum wheat variety 'Reichenbachii' is a landrace collected by Vavilov and conserved in the United States National Plant Germplasm System. The resistance of this landrace to many of the older *Pgt* races prevalent around the world has been previously reported in the past by Bechere et al. [18]. Characterization and identification of loci associated with, and molecular markers linked to resistance against the current stem rust races in this cultivar may contribute to the efficient transfer of resistance into breeding lines and may also introduce new sources of resistance to the durum germplasm pool.

The current advancement in next generation sequencing technologies with low genotyping cost is an opportunity for improving the resolution of mapping and identification of reliable markers tightly linked to QTL for stem rust resistance and other agronomically important traits [19–21]. Therefore, the objective of this study was to evaluate a durum wheat recombinant inbred line (RIL) population from a cross between the two durum wheat varieties 'DAKIYE' and 'Reichenbachii' for 1) seedling response to races JRCQC and TTRTF and field response to a bulk of the current races prevalent in Ethiopia and Kenya and 2) to conduct linkage analyses and map loci associated with seedling and field resistance to multiple *Pgt* races in this population.

## Materials and methods

### Plant material

A total of 224 $F_9$ RILs from a cross between a stem rust resistance donor parent 'Reichenbachii' (Genotype Identification (GID) 30660) and a susceptible parent 'DAKIYE' (GID 6139553; Pedigree, CMOS_3//SOMAT_4/INTER_8/3/SOOTY_9/RASCON_37/4/SFAR_1) developed by the CIMMYT durum wheat breeding program were used in this study. The RILs were evaluated along with the parents for seedling response to races JRCQC and TTRTF, and field response to multiple stem rust races in Ethiopia and Kenya for two seasons from 2019 to 2020. Races JRCQC and TTRTF were selected for seedling evaluation because of their virulence to widely deployed stem rust resistance genes/alleles (*Sr13b* and *Sr9e*) in durum wheat. The field trials were named after the country names, the seasons, and the years of evaluation. Hence, ETOS19 and ETMS19 refer to trials in Ethiopia during the off-season (January to May) and main-season (June to November) 2019, respectively while KNMS19 and KNMS20 refer to the trials in Kenya during the main-seasons (June to October) 2019 and 2020, respectively.

### Seedling evaluation

Three seeds of each line were planted in a greenhouse at Debre Zeit Agricultural Research Center (DZARC), Ethiopia in small pots (size = 8cm x 6cm x 6cm) filled with a mixture of soil, sand and farmyard manure in 2:1:1 proportion. The temperature in the greenhouse and humidity was set to a maximum of 25 ˚C and 80%, respectively under natural light. The cooling fan in the greenhouse operates when the temperature becomes above25 ˚C. Seven days old seedlings were inoculated with a urediniospore suspension of the isolates of races JRCQC (isolates Ku#3, Ku#22, Ku#30, Am#6 and BD#30 identified in 2015 and 2016 in Ethiopia), TTRTF (isolates Gi#11, Gi#31, KARC-6, DZ#15), distilled water and Tween20 as described in Woldeab et al. [22]. Inoculation was done by placing the spore suspension in a gelatin capsule connected to an inoculator and hose of an electric pump (Diaphram Pump, VACUUBRAND GMBH + CO Otto-Schott-StraBe 25 D-6980 Wertheim, W-Germany). After inoculation, the seedling pots arranged in trays were kept for about 30 minutes to allow evaporation of the tween-20. Then the trays were placed in plastic bags and a mist of water was applied using a knapsack sprayer to create a humid environment. The misted seedlings in the plastic bags were then covered with sacks to create a dark environment (a modification for incubation/dew chamber) to promote initial infection. The next morning, after at least nine hours, the bags were removed, and the seedlings were surrounded by transparent plastics to provide a humid environment. Then seedlings were transferred to another greenhouse compartment under natural light condition and scored 14 days post inoculation using the zero to four scale described in Stakman et al [23]. The zero to four Stakman scale was ";", "0", "1⁻", "1", "1⁺","2⁻", "2", "2⁺" "3⁻", "3", "3⁺", "4 –", "4", and "4⁺". This infection types (ITs) were linearized to a 0 to 9 scale to make it convenient for statistical analysis as the 0 to 4 scale has non-numeric characters ("+" and "-" to indicate the size of pustules and ";" to indicate hypersensitive response).Hence, the linearization was done as: ";" and "0" = 0, "1⁻"= 1, "1" = 2, "1⁺" = 3,"2⁻"= 4, "2" = 5, "2⁺" = 6, "3⁻"= 7, "3" = 8, "3⁺" = 9, "4" = 9 [24]. ITs of ≤ 2⁺ (≤ 6) were considered resistant and > 2⁺ (> 6) were considered susceptible.

### Field evaluation

The same RILs along with the parents were planted using a randomized incomplete block design in two replications across four testing environments (ETOS19, ETMS19, KNMS19 and KNMS20). In Debre Zeit, Ethiopia, lines were planted in a plot size of 1 m long single rows

with an inter-row spacing of 0.2 m. One moderately resistant ('Mangudo') and two susceptible ('Arendato' and 'Local Red') checks were planted after every 100 plots. Spreader rows were planted between blocks and surrounding the experimental field with a mixture of equal proportions of stem rust susceptible cultivars 'Morocco', 'PBW343', 'Digalu', and 'Arendato'. In Njoro, Kenya, lines were planted in a plot size of 0.7 m long single rows with an inter-row spacing of 0.3 m. The blocks and the experimental field were surrounded by spreader rows planted as hill plots with a mixture of stem rust susceptible cultivars 'Cacuke' and 'Robin', and six lines carrying *Sr24* (GID = 5391050, 5391052, 5391056, 5391057, 6391059, and 5391061) in equal proportions. Pathogen infection was initiated by artificial inoculation of spreader rows with a bulk of urediniospores collected from the previous field seasons of each testing environment. Spreader rows were syringe-injected with a mixture of urediniospores, distilled water and a drop of Tween20 (one drop/0.5 L) at the stage of stem elongation (~ Zadok's growth scale 31) [25]. The inoculum suspension was also sprayed twice on the spreader rows to favor uniform infection of the pathogen. The bulk of races were composed of TTKSK, TKTTF, JRCQC, TTTTF, and TRTTF in Debre Zeit, Ethiopia and TTKSK, TTKST, TTKTT, and TTTTF in Njoro, Kenya. However, these races were not the only races prevalent in the testing locations and there was variation in natural race composition in each season.

Disease severity was scored using the modified Cobb's scale (0 to 100) by estimating the proportion of stem area covered with rust pustules [26]. Infection response was scored based on the size of pustules and the amount of chlorosis or necrosis surrounding the pustules on the stem as described in Roelfs et al. [3]. The response classes are scored as '0', 'R', 'MR', 'MS' and 'S' that designate no visible infection (immune), resistant, moderately resistant, moderately susceptible and susceptible reactions, respectively. Whenever different infection responses are observed on a single genotype, combinations of response classes can be scored by taking the most frequent first followed by the less frequent response. The disease severity and response classes were combined to coefficient of infection (CI) which is the product of the disease severity and a 0 to 1 scale assigned to the response classes. The scale was specified as 0.0, 0.2, 0.4, 0.8 and 1.0 for immune, R, MR, MS, and S, respectively. In cases of combined responses per a single genotype, the mean of the scales was used to calculate CI [27]. The trials were scored at seven to 14-day intervals four times in ETOS19, three times in ETMS19 and KNMS20, and twice in KNMS19. In all trials, the last scoring was used to calculate CI and apply further statistical analyses.

## Statistical analyses of phenotypic data

The CI was used as a response variable to apply statistical analyses using R statistical software version 4.0.2 [28] and ASReml-R version 3 for spatial correction [29]. A model that resulted in the highest estimate of broad-sense heritability and/or a model with significant Wald test for fixed effects was chosen to estimate BLUPs. For ETOS19, a linear mixed model (LMM) described in Eq 1 was fitted using ASReml-R.

$$y_{ijk} = \mu + g_i + R_j + r_k + \varepsilon_{ijk} \tag{1}$$

Where: $y_{ijk}$ is the response of the $i^{th}$ genotype in the $j^{th}$ row and in the $k^{th}$ replication, $\mu$ is the overall mean response, $g_i$ is the random effect of the $i^{th}$ genotype, $R_j$ is the fixed effect of the $j^{th}$ row, $r_k$ is the random effect of the $k^{th}$ replication and $\varepsilon_{ijk}$ is the residual associated with the model. For ETMS19, a LMM described in Eq 2 was fit on the square-root transformed CI as a response variable while for KNMS19 and KNMS20 the same model was fit on the CI. The lmer() function of the R package *lme4* was used to fit the LMM [30] and the BLUPs were

estimated.

$$y_{ij} = \mu + g_i + r_j + \varepsilon_{ij} \ldots \tag{2}$$

Where: $y_{ij}$ is the response of the $i^{th}$ genotype and the $j^{th}$ replication, $g_i$ is the random effect of the $i^{th}$ genotype, $r_j$ is the random effect of the $j^{th}$ replication and $\varepsilon_{ij}$ is the residual.

For seedling responses of lines to races JRCQC and TTRTF, the linearized scale of the ITs (0 to 9) was considered for further statistical analysis. The model described in Eq 2 was fit to estimate the variance components. The broad-sense heritability was calculated by applying Eq 3 on the estimated variance components from the respective models fit on the data [31].

$$H^2 = V_g / V_p \tag{3}$$

Where $H^2$ is the broad-sense heritability, $V_g$ is the variance due to the genotype or line, $V_p$ is the phenotypic variance, $V_p = V_g + V_e$, where $V_e$ is the residual variance. The presence of transgressive segregants i.e. lines with extreme phenotypes than either of the parents were assessed and t.test was conducted to determine the statistical significance of the difference between the parent phenotype and the line with extreme phenotype than the parents.

## Genotyping and SNP calling

Leaf tissue was sampled into 1.1 ml strip tubes arranged in 96-well plate format. Tissue was collected from seedlings of the two parents and each RIL (226 lines in total) grown in a greenhouse at CIMMYT, Mexico. Samples were frozen at -80˚C for three hours and subsequently lyophilized for 48 h. Lyophilized leaf samples were ground using a GenoGrinder 2010 (SPEX, SamplePrep, Metuchen, NJ) for 2–3 minutes by placing stainless steel balls for pulverization. Genomic DNA was extracted using the modified cetyl trimethylammonium bromide protocol as described in Dreisigacker et al. [32] and shipped to the USDA-ARS Eastern Regional Small Grains Genotyping Lab. in Raleigh, NC for genotyping. The extracted DNA was quantified using Quant-iT PicoGreen reagents (ThermoFisher Scientific, Waltham, MA) in flat bottom plates on a BMGLabTech (Ortenberg, Germany) PHERAstar Plus plate reader with MARS software. Genomic DNA was then subjected to a *PstI-MspI* double restriction digest, followed by sequencing adapter ligation and library preparation as described in Poland et al. [33]. Single-ended 100bp read length Illumina (San Diego, CA) sequencing was performed on a Novaseq 6000 instrument. Single Nucleotide Polymorphism (SNP) genotypes were called using the TASSEL-GBS v2 pipeline [33] in TASSEL v5 [34]. Reads were aligned to the durum wheat cultivar 'Svevo' v1.0 reference sequence [35] using the Burrows-Wheeler Aligner (BWA) v0.7.17 [36]. Reads that aligned to unordered contigs were discarded.

**Genotype data filtering and linkage map construction.** SNP markers containing greater than 20% missing data or greater than 10% heterozygous calls were removed. In addition, all but one SNP was removed from clusters of SNPs in perfect linkage disequilibrium (LD, $r^2 = 1$) with one another. Following filtering, 7418 SNPs in 201 lines remained for further analysis. The SNP markers were converted to the ABH-genotype format (A = homozygous for allele inherited from susceptible parent; B = homozygous for allele inherited from resistant parent and H = heterozygous) using TASSEL [34]. On conversion of the SNP markers to ABH-genotypes using the *read.cross*() function of R/*qtl* package v1.46–2 [36], 929 marker genotypes for 198 lines were generated for the next filtering steps. Heterozygous calls from the *read.cross*() output were replaced by missing data that was imputed using the R/*qtl fill.geno*() function. In the *fill.geno*() function, the "*argmax*" method that uses the most likely sequence given the observed data was applied.

The data was further diagnosed for the presence of outlier genotypes for each line and marker, excessive proportion of shared alleles between lines, marker genotypes with

segregation distortions and genotyping errors, markers with misaligned positions, and SNP markers and lines with excess double crossover/crossover counts. Chi-square tests were conducted to evaluate segregation distortion (deviation from the expected 1:1 ratio) at a Boneferroni threshold for multiple test correction (*P-value* < 5.38e-05). Linkage groups were formed at minimum LOD score value of six and maximum recombination frequency of 35%. Lines with more than 95% shared alleles, switched markers to different positions/linkage groups, marker genotypes with double crossover counts above 10 and lines with crossover counts greater than 60 were discarded from the dataset.

The genetic map was estimated at each filtering step. The R/*qtl ripple*() function package was used for the likelihood ratio test that assesses all possible permutations of marker orders and recombination frequencies. Markers were assigned genetic distances (centiMorgans) using the Kosambi mapping function [37]. The marker order with the highest LOD score and the shortest possible length was chosen for each chromosome. The final genetic map was graphically represented using 843 quality markers for 175 RILs using the R/*qtl plotMap*() function.

**QTL analyses.**   Before conducting QTL analyses, QTL genotype probabilities were calculated using the R/*qtl calc.genoprob*() function at a step of 2 cM with an assumed genotyping error rate of $1.0e^{-4}$ and using the Kosambi mapping function [37]. QTL analysis was conducted using composite interval mapping (CIM) [38,39] and the Haley-Knott regression method [40]. The BLUPs estimated from the LMM fitted on field response data and the mean linearized scale of the two replications for seedling response were fit as response variables for the QTL analyses. No significant regions were identified using Logarithm of odds (LOD) thresholds identified by 1000 permutation tests at an experiment-wise $\alpha = 0.05$ and $\alpha = 0.10$ and a window size of 10 cM for the field evaluation. Therefore, a LOD score of 2.5 was set as a threshold to declare the identification of significant QTL. For the seedling response, LOD thresholds were estimated using 1000 permutation tests at an experiment-wise $\alpha = 0.05$.

Markers flanking the QTL were identified using R/*qtl lodint*() function that calculates the 1.5 LOD intervals. The effects of QTL on the phenotype and the percentage of variance in the phenotype explained by the QTL were identified by fitting a linear model using the R/*qtl fitqtl*() function. The donor of the identified QTL for resistance among the two parents and the QTL effect was visualized using the R/*qtl effectplot*() function. Then the presence of QTL-by-environment interaction was examined by fitting a linear model using the BLUPs as a response variable and the QTL, environment, and QTL-by-environment interaction as explanatory variables for the field evaluation.

## Results

### Phenotypic data analyses

**Seedling evaluation.**   The distributions of the linearized scale of seedling IT for response to races JRCQC and TTRTF were skewed towards the resistance score (Fig 1). The broad-sense heritability was 0.80 and 0.76 for seedling responses to races JRCQC and TTRTF, respectively (Table 1). The mean IT of the resistant parent was 0 for response to both races and that of the susceptible parent was 6 and 8.5 for response to races JRCQC and TTRTF, respectively. Among the lines evaluated, 41% were resistant (IT $\leq 2^{+}$ or $\leq 6$) to race JRCQC and 29% were resistant to race TTRTF.

### Field evaluation

The frequency distribution of the CI of the RIL population for response to a bulk of *Pgt* races was close to normal for ETOS19 and KNMS19 but skewed towards the resistant score for

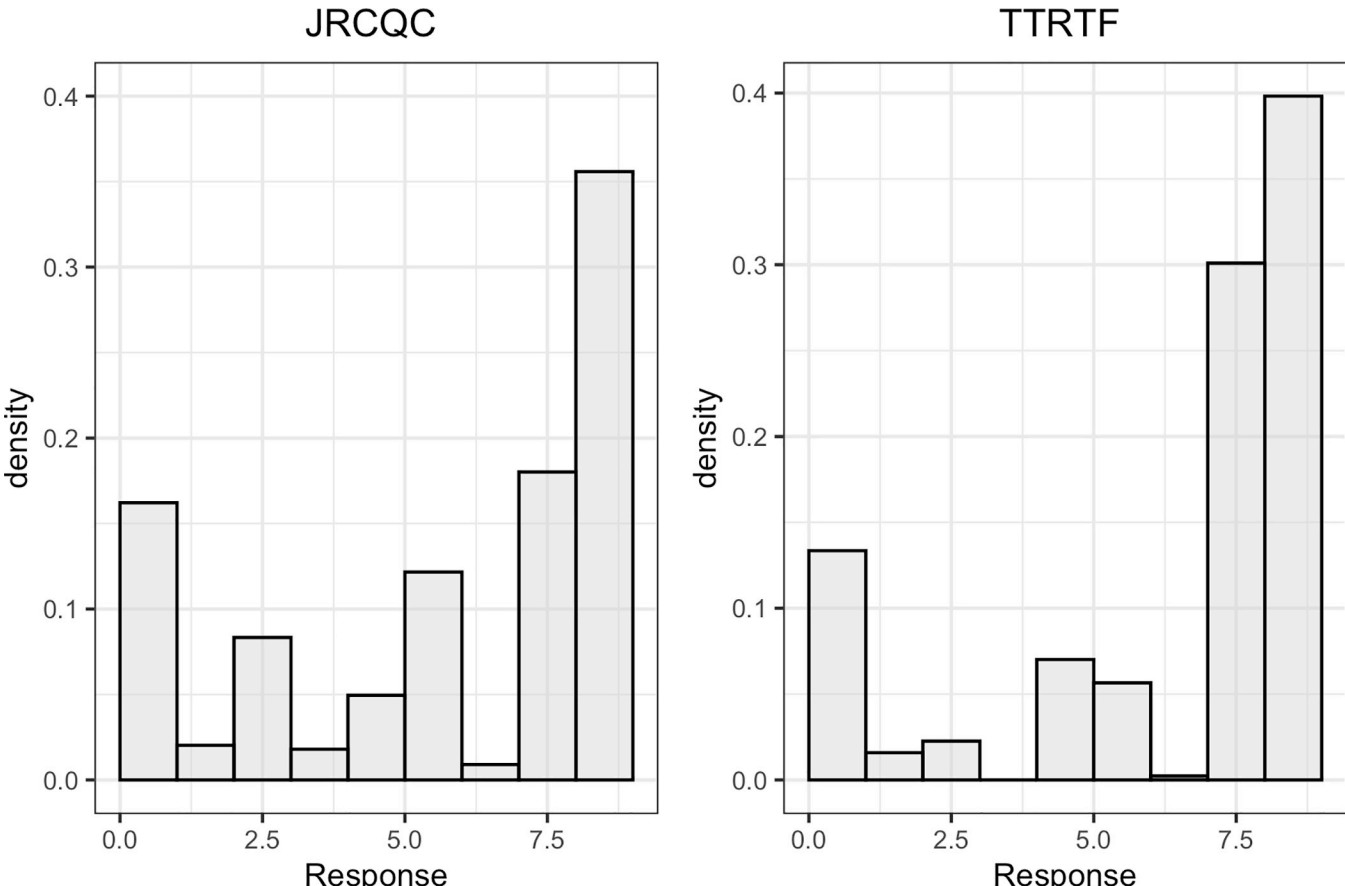

**Fig 1. Distribution of seedling responses of the RIL population derived from 'Reichenbachii' /DAKIYE to races JRCQC and TTRTF.**

ETMS19 and was nearly a bimodal for KNMS20 (Fig 2). The broad-sense heritabilities estimated from the variance components were 0.58, 0.62, 0.85 and 0.84 in ETMS19, ETOS19, KNMS19 and KNMS20, respectively. The mean CI of the resistant parent ('Reichenbachii') ranged from 0 in KNMS20 to 6 in ETMS19 while that of the susceptible parent varied from 35.6 in ETMS19 to 85 in KNMS20 (Table 2).

Assuming a disease score of 30MRMS or lower as resistant in the field (CI = 30 x 0.6 = 18), 7%, 33%, 44.7% and 38.7% were resistant (CI ≤ 18) in ETOS19, ETMS19, KNMS19 and KNMS20, respectively. The proportion of susceptible lines ranged from 55.3% in KNMS19 to

**Table 1. Mean, genetic variance and broad-sense heritability of seedling infection response of the RIL population against races JRCQC and TTRTF.**

| Statistic | JRCQC | TTRTF |
|---|---|---|
| Overall mean Infection Type (IT) | 6.05 | 6.8 |
| Mean of resistant parent | 0 | 0 |
| Mean of susceptible parent | 6.0 | 8.5 |
| Percent of resistant lines | 41.4 | 28.9 |
| Percent of susceptible lines | 58.6 | 71.0 |
| Genetic variance | 8.82 | 7.17 |
| Heritability | 0.80 | 0.76 |

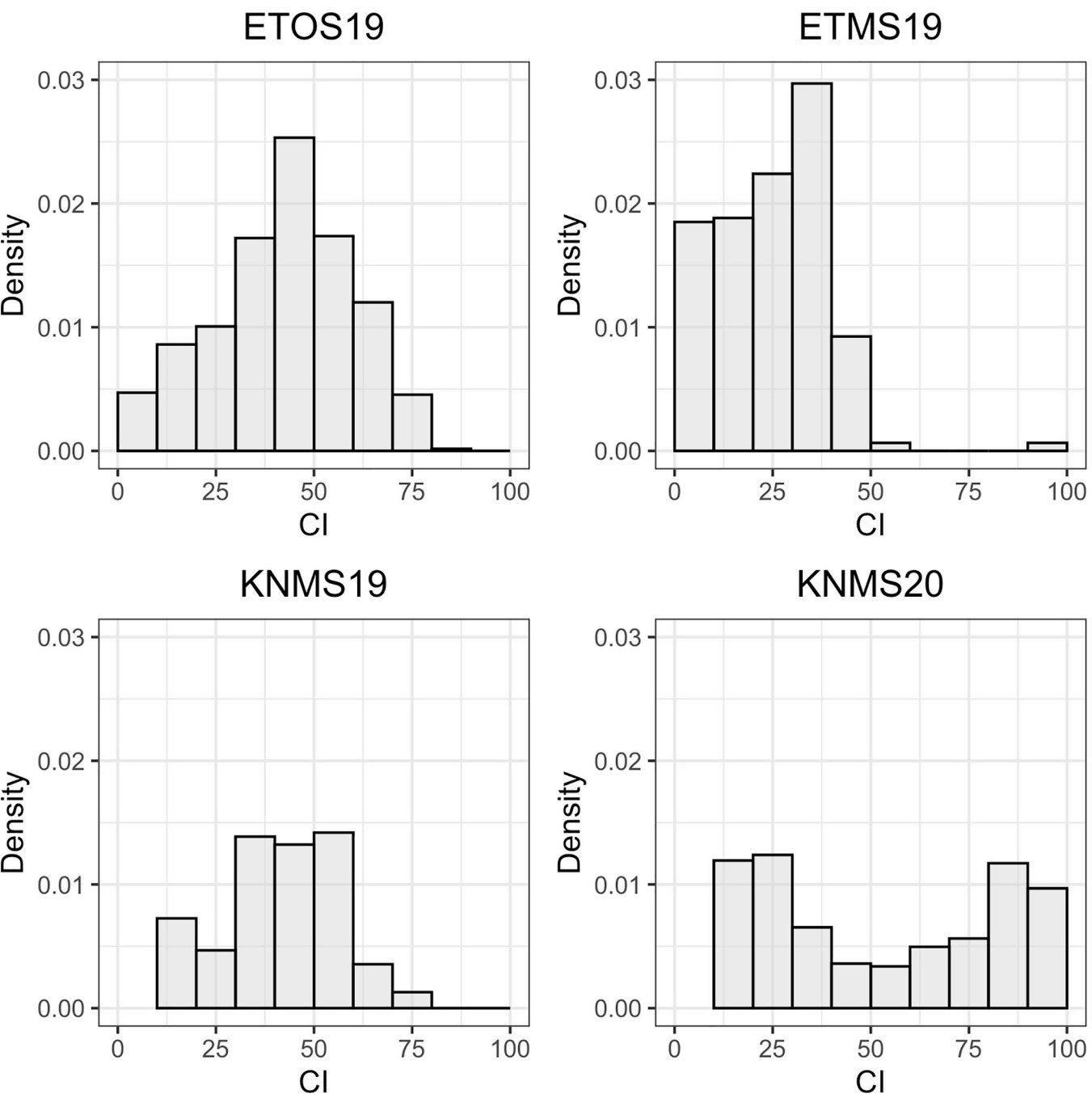

**Fig 2. Distribution of CI of field responses of the RIL population derived from 'Reichenbachii' /DAKIYE cross in four testing environments.**

93% in ETOS19. Among the total number of resistant lines in each environment, none were transgressive segregants for resistance in ETOS19 and KNMS20 but 1% (one line, GID 8600910) and 15.7% (21 lines) were transgressive segregants for resistance in ETMS19 and KNMS19, respectively (Table 3). From the 21 transgressive segregants for resistance against races in Kenya, only four lines had marker data. Among the four lines, line GID 8600960 was a non-parental type (the alleles were different from the parents).

**Table 2. Mean, genetic variance and broad-sense heritability of CI of RIL population across four testing environments.**

| Statistic | ETOS19 | ETMS19 | KNMS19 | KNMS20 |
|---|---|---|---|---|
| Overall mean | 43.9 | 26.3 | 25.4 | 39.9 |
| Mean of resistant parent | 3.6 | 6.0 | 0.6 | 0.0 |
| Mean of susceptible parent | 67.5 | 35.6 | 47.3 | 85.0 |
| Genetic variance | 207 | 2.08 | 448.5 | 1090 |
| Heritability | 0.62 | 0.58 | 0.85 | 0.84 |

## Data filtering and linkage map construction

Several steps of filtering were undertaken before construction of the genetic linkage map and QTL analyses. The heatmap of the marker data before imputation, and after imputation and filtering is presented in S1A and S1B Fig. No outlier line and marker genotypes were detected after imputation. Lines were compared for their shared proportion of alleles and 20 lines with > 95% shared alleles were discarded (S2 Fig). On a chi-square test of the deviation from a 1:1 segregation of marker genotypes, 47 markers showed significant segregation distortion at Boneferroni threshold (*P-value* < 5.38e-05) (S1 Table) and these markers were also discarded.

Misaligned markers that mapped to a different linkage group and markers with switched alleles were omitted from the dataset based on the recombination fraction and LOD score heatmap (Fig 3). Twenty-one markers with double crossover counts above 10 and three lines with marker crossover counts ≥ 60 were removed from the dataset. SNP markers were tested for the presence of genotyping errors with an assumed error rate of 0.01 and no marker with genotyping error above the cutoff (error LOD score = 4) was identified.

The 843 markers were distributed across 13 linkage groups representing all chromosomes of durum wheat except chromosome 7A (Figs 4 and S3). These markers covered 1458.1 cM of the genome with an average interval of 1.73 cM. The B sub-genome had a larger number of SNPs (535) than the A sub-genome (308) (S3 Fig). The A sub-genome covered 674.4 cM with an average interval of 2.19 cM while the B sub-genome covered 783.7 cM with an average interval of 1.23 cM. Chromosome 3B carried the largest number of SNPs covering a genetic distance of 207.3 cM followed by chromosome 7B (171.2 cM). SNPs on chromosome 7A were dropped during the filtering steps and chromosomes 2B and 4A had the lowest marker coverage with 4.4 cM and 12.6 cM, respectively (Figs 4 and S3).

## QTL mapping

**Seedling evaluation.** Composite interval mapping identified a single QTL associated with seedling resistance to races JRCQC and TTRTF on chromosome 3B at 64 and 66 cM, respectively (Table 4). Sub-threshold QTL peaks were identified on chromosomes 3A for both races and on 7B for race JRCQC (Fig 5). The LOD score of the QTL on 3B (*QSr.cnl-3B;* named

**Table 3. Percentage of resistant, susceptible, and transgressive segregants of RILs evaluated for response to multiple stem rust races across four testing environments.**

| Environment | Percent resistant | Percent susceptible | Percent transgressive segregants | |
|---|---|---|---|---|
| | | | Resistant | Susceptible |
| ETOS19 | 7 | 93 | 0 | 19 |
| ETMS19 | 33 | 67 | 1 | 86 |
| KNMS19 | 44.7 | 55.3 | 15.7 | 66 |
| KNMS20 | 38.7 | 61.3 | 0 | 35 |

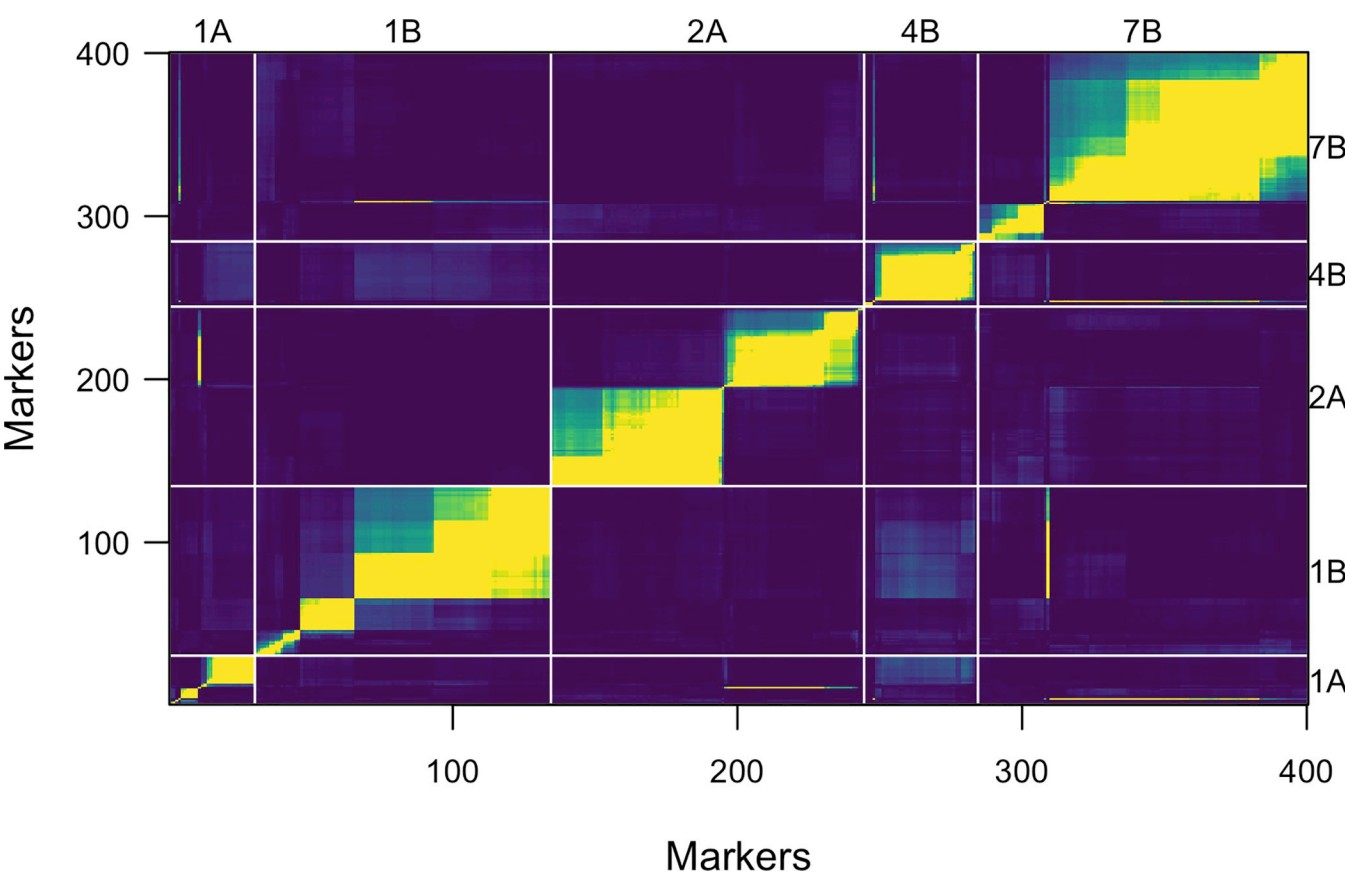

**Fig 3.** Heatmap of recombination fraction (upper left triangle) and LOD score (lower right triangle) of selected chromosomal regions with misaligned markers indicated by yellow strip on the blue background.

according to McIntosh et al. [41]) was 5.26 for seedling response to race JRCQC and 3.33 for race TTRTF. This QTL explained 11.8% and 6.5% of the phenotypic variation for seedling response to races JRCQC and TTRTF, respectively (Fig 6). The peak marker for the QTL *Sr. cnl-3B* was mapped between S3B_91123277 (4 cM away) and S3B_196992709 (3 cM away) for response to race JRCQC and between markers S3B_75830161 (6 cM away) and S3B_195726515 (4 cM) for race TTRTF (Table 4).

### Field evaluation

Composite interval mapping detected three significant QTL (one per testing environment) on chromosomes 3B, 4B and 7B associated with field resistance to multiple stem rust races in Ethiopia and Kenya (Table 5). Sub-threshold QTL peaks were observed on chromosomes 1A and 5A (Fig 7). The LOD scores of the QTL identified ranged from 2.52 to 4.29 (Table 5). The QTL *QSr.cnl-3B* located at 66 cM and 67 cM on chromosome 3B was significant in two (ETMS19 and KNMS19).of the four environments (Table 5) This QTL explained 6.4% and 15.3% of the phenotypic variation in ETMS19 and KNMS19, respectively. The additive effect was -7.85 in KNMS19 and -0.27 in ETMS19 (S3 Table). The peak marker for *QSr.cnl-3B* in ETMS19 (S3B_166187578) was mapped between markers S3B_91123277 (5.1 cM away) and S3B_259053349 (4.8 cM away). In KNMS19, the peak marker for the same QTL was c3B.loc66 and was mapped between markers S3B_343854 and S3B_196992709, 66 cM and 3.5 cM away, respectively. A QTL on chromosome 4B (*QSr.cnl-4B*) identified in ETOS19 explained 4.7% of

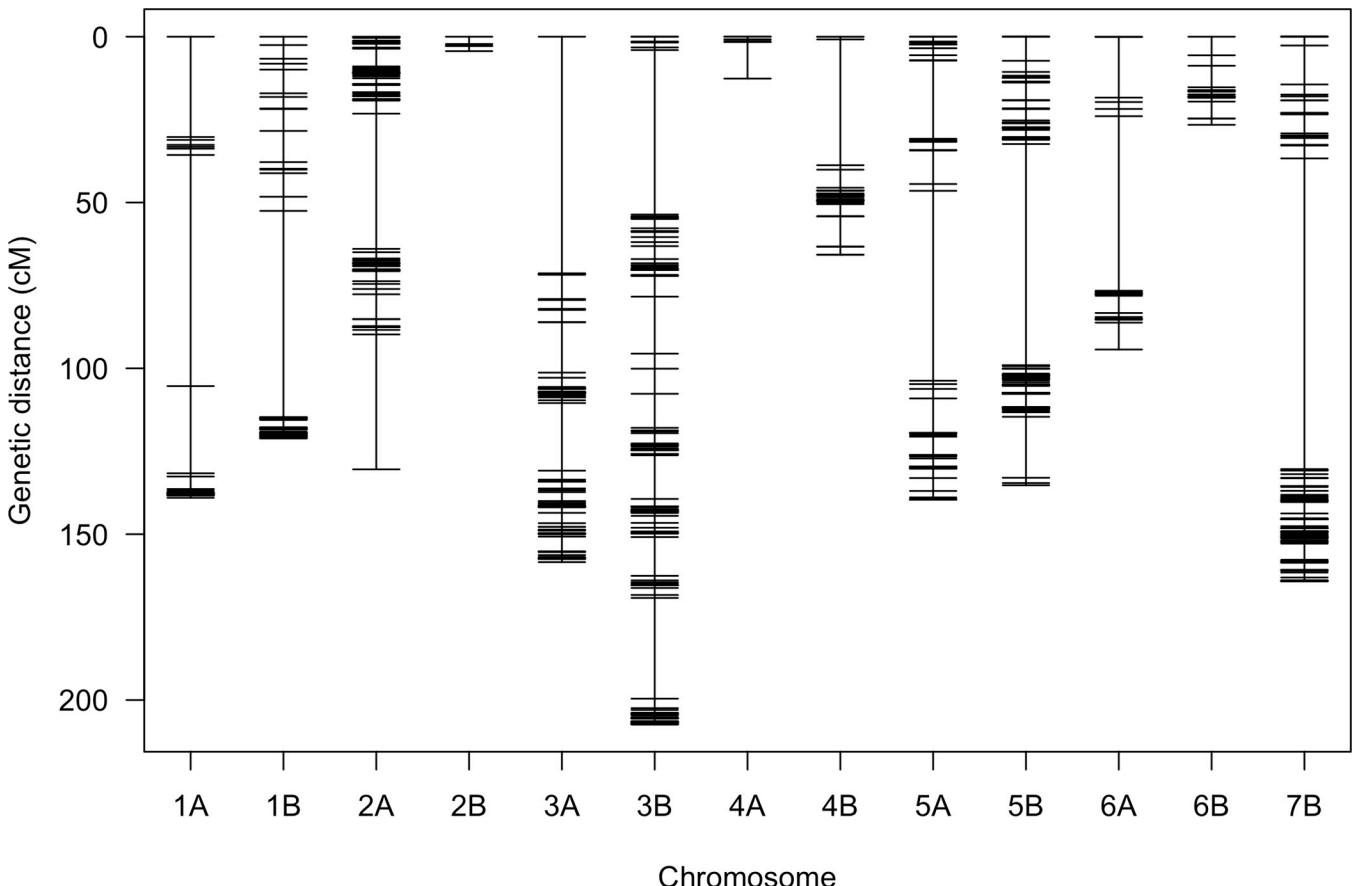

**Fig 4. Genetic linkage map constructed from SNP markers derived from genotyping-by-sequencing in a recombinant inbred line population from a cross between Reichenbachii and DAKIYE.**

the phenotypic variation (additive effect = -2.75) (Tables 5 and S3). The peak marker of QTL *QSr.cnl-4B* (S4B_524068577) was mapped 37.9 cM away from marker S4B_8811137 and 6.8 cM away from marker S4B_550731907 (Table 5).

In KNMS20, a QTL was identified at 143 cM on chromosome 7B (*QSr.cnl-7B*). A marker linked to this QTL (c7B.loc136) was located 2.9 cM and 7.7 cM away from markers S7B_677752911 and S7B_688049535, respectively. QTL *QSr.cnl-7B* explained 7.2% of the phenotypic variation (additive effect = -8.64) for field resistance to multiple stem rust races in KNMS20 (Tables 5 and S3). All three QTL were contributed by the resistant parent, 'Reichenbachii' and the QTL on chromosome 3B (*QSr.cnl-3B*) explained the highest percentage of the

**Table 4. QTL identified using composite interval mapping for seedling response to races JRCQC and TTRTF.**

| Race | QTL name | SNP.ID | Flanking markers | | Pos[a] (cM) | LOD | R[2b] |
|---|---|---|---|---|---|---|---|
| | | | Left | Right | | | |
| JRCQC | *QSr.cnl-3B* | c3B.loc66 | S3B_91123277 | S3B_196992709 | 66.0 | 5.26 | 11.8 |
| TTRTF | *QSr.cnl-3B* | c3B.loc64 | S3B_75830161 | S3B_195726515 | 64.0 | 3.33 | 6.5 |

[a] Position in cM.

[b] Values indicate the percentage of phenotypic variance explained by the QTL.

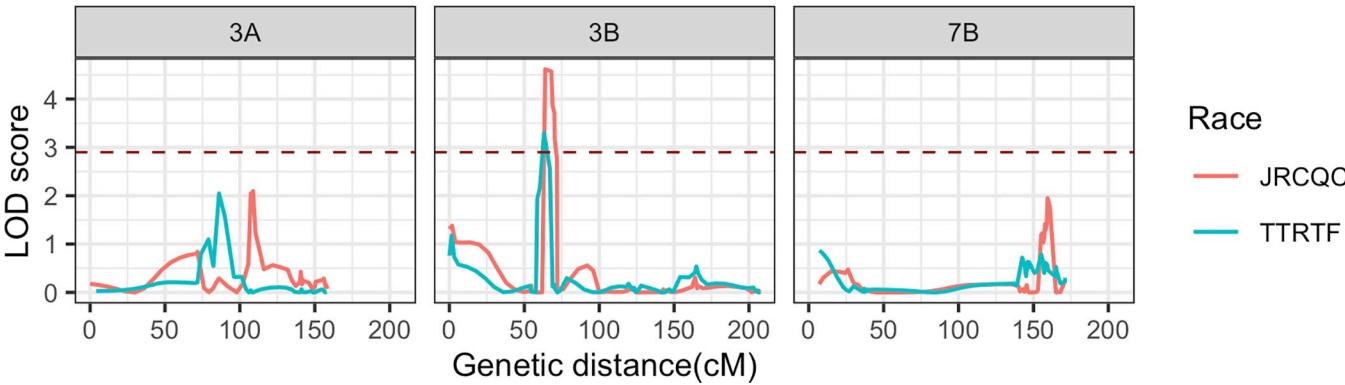

**Fig 5. LOD score curves of selected chromosomes from composite interval mapping results for seedling responses to races JRCQC and TTRTF, the brown dotted line indicates the LOD threshold (2.9).**

**Fig 6. Effects of QTL on the response of RILs to races JRCQC and TTRTF, the A allele was from the susceptible parent ('DAKIYE') and the B allele was from the resistant parent ('Reichenbachii').**

**Table 5. QTL identified using composite interval mapping for field response to bulk of *Pgt* races across four testing environments.**

| Env.[a] | QTL name | SNP.ID | FM[b] | | Pos[c] (cM) | LOD | R²[d] |
|---|---|---|---|---|---|---|---|
| | | | Left | Right | | | |
| ETOS19 | *QSr.cnl-4B* | S4B_524068577 | S4B_8811137 | S4B_550731907 | 38.7 | 2.5 | 4.7 |
| ETMS19 | *QSr.cnl-3B* | S3B_166187578 | S3B_91123277 | 3B_259053349 | 67.0 | 2.84 | 6.4 |
| KNMS19 | *QSr.cnl-3B* | c3B.loc66 | S3B_343854 | S3B_196992709 | 66.0 | 4.3 | 15.3 |
| KNMS20 | *QSr.cnl-7B* | c7B.loc136 | S7B_677752911 | 7B_688049535 | 143.0 | 2.7 | 7.2 |

[a] Environment, ETOS19 and ETMS19 = Ethiopia off-season2019 and main season 2019, respectively; KNMS19 and KNMS20 = Kenya main-season 2019 and 2020, respectively.

[b] Flanking markers.

[c] Position in cM.

[d] Values indicate the percentage of phenotypic variance explained by the QTL.

phenotypic variation in KNMS19 (15.3%) followed by the 7B locus (*QSr.cnl-7B*) in KNMS20 (7.2%) (Fig 8 and Table 5). The QTL by environment interaction was significant for *QSr.cnl-3B* (*P-value* = 6.705e$^{-05}$) and *QSr.cnl-7B* (*P-value* = 3.489e$^{-04}$) but not significant for *QSr.cnl-4B* (*P-value* = 0.10666) (S2 Table).

## Discussion

We mapped QTL associated with seedling and field resistance in a RIL population derived from a cross between a resistant parent 'Reichenbachii' and a susceptible parent 'DAKIYE' to races JRCQC and TTRTF at the seedling stage and to a bulk of multiple *Pgt* races prevalent in Ethiopia and Kenya.

### Seedling evaluation

The distribution of seedling responses of RIL population to races JRCQC and TTRTF appears skewed to the resistance score (Fig 1). However, the higher frequency of seedling resistant lines to race JRCQC than race TTRTF may indicate that some of the lines resistant to the former race could be susceptible to the latter (Table 1). The large variance in seedling response to both races explained by the RILs (80% for JRCQC and 76% for TTRTF) agrees with the qualitative nature of seedling resistance to the two durum virulent races (Table 1).

### Field evaluation

For the field response to a bulk of multiple races, the RIL population responded differently in the four testing environments. The differences in the distribution of the CI of the RILs across the testing environments suggested that there was variation in race composition. The overall mean CI (43.9) and the percentage of susceptible lines (93%) were the highest in ETOS19 indicating that there may be a high frequency of virulent races in this nursery (Table 3). This can be explained by a higher disease pressure favored by the warm and humid environment in the off-season in comparison to the main season resulting in a better screening environment. The near bimodal distribution in KNMS20 may suggest that there was a single resistance gene segregating in the population in this environment (Fig 2). Evaluation of the transgressive segregants identified in ETMS19 and KNMS19 against multiple races at the seedling and adult plant stage may help to understand the type of resistance (Table 3).

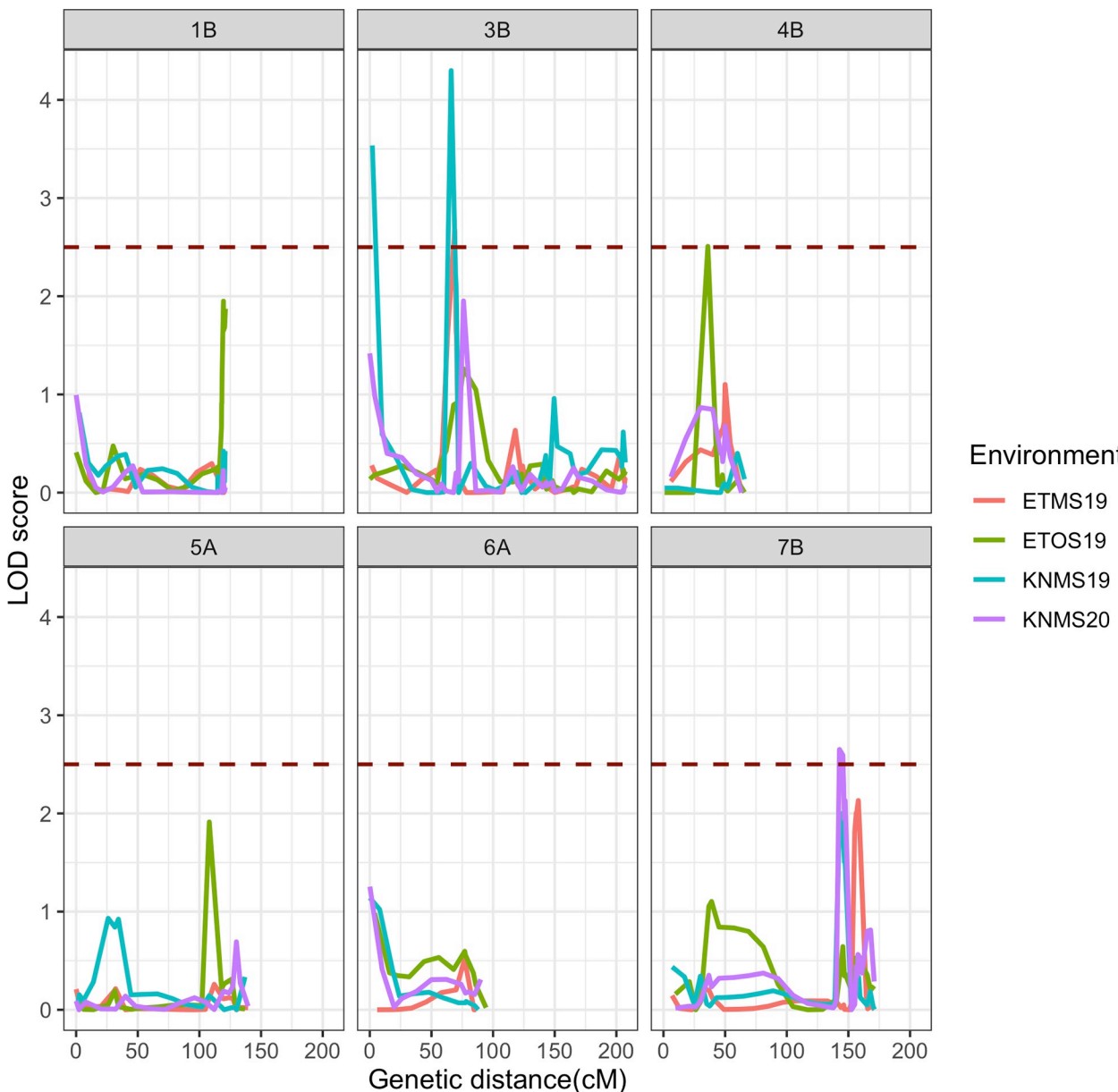

**Fig 7. LOD score curves of selected chromosomes from composite interval mapping results across the four testing environments, the brown dotted line indicates the LOD threshold (2.5).**

### Data filtering and linkage map construction

Several low-quality markers were discarded after applying different filtering criteria. To minimize the loss of information, imputation on marker data with a high proportion of missing data ($\leq 50\%$) was attempted. However, this resulted in an overestimation of recombination and extended genetic map distances were observed in this population. Therefore, we used an imputed dataset with less missing data (20%) and the genetic map presented in Fig 4 was generated. This genetic map was improved but still had uneven distribution of markers in most of the linkage groups. This was due to the removal of 6,575 low-quality markers leaving 843 markers after filtering as indicated in the methods and results sections. The removal of many

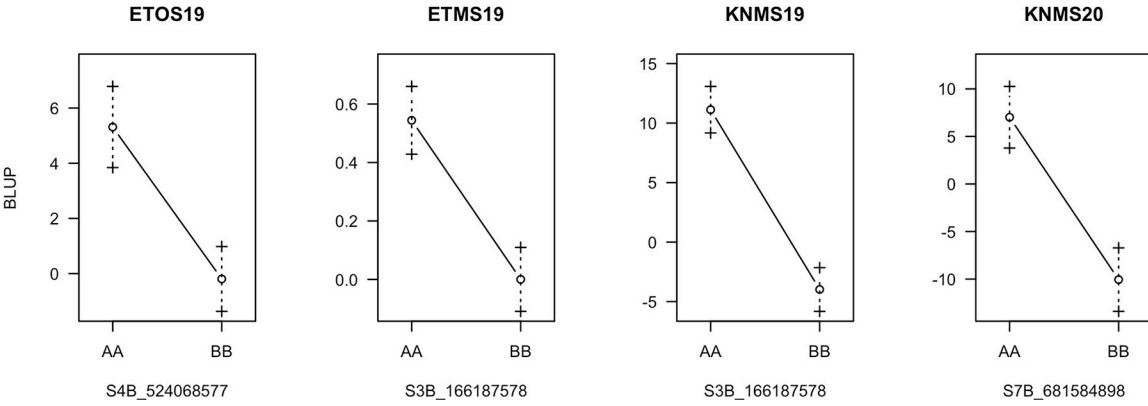

**Fig 8. Effects of QTL on the response of RILs across the testing environments, the A allele was from the susceptible parent ('DAKIYE') and the B allele was from the resistant parent ('Reichenbachii').**

markers may also affect estimation of genetic distance, resolution of mapping and the power to identify QTL however, filtering out low quality markers is a critical step before construction of a linkage map and QTL analysis. The causes for the low-quality markers and difficulty in constructing the map are unknown. A selection from the 'Reichenbachii' landrace variety was the resistant parent and its origin is unknown. Variety 'Reichenbachii', the oldest accession in the USDA National Germplasm Repository, was collected in 1940 from Portugal. Genome structural rearrangements of alien introgressions can cause abnormal segregation and additional cytological investigations will be required to resolve this question.

## QTL mapping for seedling and field response to *Pgt* races

The continuous distribution of disease responses across the testing environments except in KNMS20 indicated the presence of more than one locus responsible for resistance to multiple races of stem rust in this RIL population (Fig 2). However, only one significant QTL was detected per environment. This could be due to the QTL by environment interaction (S2 Table) and the low marker density in most of the linkage groups which reduced the power to detect additional QTL (Figs 4 and S3).

QTL *QSr.cnl-3B* (located at 64, 66 and 67 cM) was associated with seedling resistance to races JRCQC, TTRTF, and field resistance to a bulk of stem rust races in ETMS19 and KNMS19 (Tables 4 and 5). The short arm of chromosome 3B harbors the known adult plant resistance gene *Sr2* originating from emmer wheat (*Triticum dicoccum*). This gene is characterized by slow rusting and the pseudo black chaff trait. The pseudo black chaff trait and adult plant resistance cannot be identified at the seedling stage and is not always expressed in the adult plant [42]. *Sr2* is recessive McIntosh et al. [42], and Bechere et al. [18] reported that the resistance in 'Reichenbachii' was also recessive inheritance. However, the QTL we identified was associated with seedling and field resistance and it was 55 cM to 67 cM away from six markers reported by Bajgain et al. [43], marker cs*Sr2* reported by Mago et al. [44] and Yu et al. [45] representing the *Sr2* locus. Thus, the region identified on chromosome 3B is unlikely to be the *Sr2* locus. The all-stage resistance gene *Sr12* is also located on chromosome 3B. Rouse et al. [46] reported that *Sr12* is involved in adult plant resistance as a result of complementary epistasis with other resistance genes. Markers *IWA4195*, *IWA4630*, *IWA4235*, *IWA3218*, *IWA610*, *IWA611* reported by Chao et al. [47] for seedling resistance of diverse durum wheat lines to race BCCBC that map the region of *Sr12* were located between 87 cM to 88 cM and this locus is 21 cM to 24 cM away from *QSr.cnl-3B* (Tables 4 and 5). Moreover, the monogenic

differential line carrying *Sr12* showed a susceptible response to race TTKSK at the seedling stage (3⁺) and in field evaluation against races in Kenya (Disease score from 60S to 80S) where the QTL is detected [12]. Hence, the locus detected on chromosome 3B is unlikely to be *Sr12*. The peak markers of *QSr.cnl-3B* (c3B.loc64 for race TTRTF, c3B.loc66 for JRCQC and KNMS19, S3B_166187578 for ETMS19) collocate (0.5 cM to 1.5 cM away) with QTL tagging markers *IWB24497*, *IWB30621*, *IWB42046*, *IWB4823*, *IWB56471*, *IWB61425* (67.5 cM) reported by Bajgain et al. [48] for seedling resistance of diverse spring wheat lines to Ug99. It is known that the Ug99 group of races are predominant in Kenya. The QTL on chromosome 3B was identified in the field and in the greenhouse for resistance to races JRCQC and TTRTF. Therefore, it is likely that this QTL is the same or tightly linked the QTL identified by Bajgain et al. [48].

One of the flanking markers of a QTL in PBW343/Kingbird population reported by Bhavani et al. [49] (*tPt-0602*) and a QTL in a durum wheat diversity panel reported by Letta et al. (2013) (*wPt-8543*) are further away from QTL *QSr.cnl-4B* (132 Mb to 153 Mb away) identified for field resistance in ETOS19 (Table 5). QTL *QSr.cnl-4B* is close to *QSr.umn-4B.2* linked to marker *wsnp_Ku_c8075_13785546* (4.4 cM away) reported by Bajgain et al. [43] for adult plant resistance of the RBO7/MNO6113-8 RIL population to *Pgt* races in St. Paul, Minnesota (Table 5). This QTL showed the smallest effect ($R^2 = 4.74$) and may identify the same region as the one reported by Bajgain et al. [43]. Field screening of the RILs against races in Minnesota is needed to understand whether the same QTL is effective against races in Minnesota and Ethiopia. If this region is novel, it will be useful for breeding durum wheat resistant to virulent races predominant in Ethiopia.

QTL *QSr.cnl-7B* identified in KNMS20 is close to the *Sr17* locus identified by markers *wPt-1715*, *wPt-4298*, *wPt-7991*, *wPt-4045* reported by Letta et al. [50] for field resistance of diverse durum wheat lines against races in Ethiopia. Marker *wmc517* reported by Letta et al.[51] tagging the *Sr17* locus for seedling resistance against races TTTTF and TTKSK is 5 cM away from one of the flanking markers of *QSr.cnl-7B* (S7B_688049535). Although the marker platform is different, the QTL we identified on chromosome 7B could be the *Sr17* locus and the region was also physically close (1Mb to 4Mb away) to the markers linked to the *Sr17* locus.

None of the QTL identified except the *QSr.cnl-3B* were consistent for the races evaluated and across the evaluation environments. This suggests that this QTL is reliable and effective at all growth stages. The QTL effect on disease reduction was larger against races in Kenya (additive effect = -7.85) than races in Ethiopia (additive effect = -0.27) which could indicate the presence of more virulent races in Ethiopia than in Kenya (S2 and S3 Tables). The interaction of the QTL with multiple-races prevalent in the testing environments could be another reason for the lower effect of the QTL on the field response (S3 Table). Therefore, evaluation of the RIL population against the responses to single races in the field may elucidate the real effects of these QTL on the response. Seven lines (GID 8600810, 8600943, 8600964, 8600976, 8601029, 8601043, 8601058) can be used as parents in crossing for stem rust resistance as they were consistently resistant in all testing environments and carried two to three of the resistant alleles at the identified loci.

## Conclusion

In summary, the three QTL contributed by 'Reichenbachii' (the resistance donor parent) identified in this study were previously reported in common wheat. As the QTL effect on the response in the current study was generally small, evaluation of the RIL population against single races in the field may uncover the specific effects of the QTLs on the response. The marker linked to the QTL on chromosome 3B can be utilized in marker-assisted selection since this

QTL was consistent across environments and at all growth stages. RILs resistant in all the four environments in the field can be used as parents in crossing. As most of the RILs were very tall and susceptible to lodging, evaluation of the RILs for the presence of other undesirable traits linked to the QTL that could potentially be transferred to elite lines prior to using this parent in resistance breeding will be needed. The power to identify additional QTL in this RIL population will be limited by abnormal segregation resulting in the removal of several low-quality markers and a cytological study is needed to uncover the presence of chromosomal rearrangements as the donor parent is a landrace from older collections. Since many of the commercially deployed stem rust resistance genes in durum wheat are qualitative including those identified in this study, evaluation of large numbers of durum lines to identify sources of durable adult plant resistance to stem rust is crucial in the future resistance breeding of durum wheat against stem rust.

## Supporting information

**S1 Fig. Distribution of alleles from the susceptible parent (DAKIYE) coded as A and the resistant parent (Reichenbachii) coded as B.** Red represents the allele from the susceptible parent and blue represents the allele from the resistant parent. The white spaces in S1A Fig were missing data and S1B Fig was after imputation and filtering. R-code adapted from Hussain et al. [52].
(TIFF)

**S2 Fig. Proportion of shared alleles between RILs from 'DAKIYE'/'Reichenbachii' cross.**
(TIFF)

**S3 Fig. Distribution of SNP markers of RILs derived from genotyping-by-sequencing across linkage groups/chromosomes.**
(TIFF)

**S1 Table. Lists of marker genotypes with significant segregation distortion at Bonefferroni threshold.**
(DOCX)

**S2 Table. Summary of analysis of variance and QTL by environment interaction.**
(DOCX)

**S3 Table. Summary additive effect of QTL identified for seedling and field response to stem rust.**
(DOCX)

## Acknowledgments

The authors are thankful to the EIAR for the leave of absence and other supports to the student during the study. We are also grateful to collaborators from CIMMYT, EIAR, KALRO, and USDA-ARS, Eastern Regional Small Grains Genotyping Lab, Raleigh, NC. The authors also would like to extend sincere appreciation to Ashenafi Gemechu, Bekele Abiyo, Bedada Girma and Ayele Badebo for processing the domestic quarantine after seed import.

## Author Contributions

**Conceptualization:** Shitaye Homma Megerssa.

**Data curation:** Shitaye Homma Megerssa.

**Formal analysis:** Shitaye Homma Megerssa.

**Funding acquisition:** Maricelis Acevedo, Mark Earl Sorrells.

**Investigation:** Shitaye Homma Megerssa.

**Methodology:** Shitaye Homma Megerssa, Karim Ammar, Susanne Dreisigacker, Mandeep Randhawa, Gina Brown-Guedira, Brian Ward.

**Project administration:** Shitaye Homma Megerssa, Mark Earl Sorrells.

**Resources:** Karim Ammar, Maricelis Acevedo, Mark Earl Sorrells.

**Software:** Shitaye Homma Megerssa, Brian Ward.

**Supervision:** Maricelis Acevedo, Gary Carlton Bergstrom, Mark Earl Sorrells.

**Validation:** Shitaye Homma Megerssa.

**Visualization:** Shitaye Homma Megerssa.

**Writing – original draft:** Shitaye Homma Megerssa.

**Writing – review & editing:** Karim Ammar, Maricelis Acevedo, Gary Carlton Bergstrom, Susanne Dreisigacker, Mark Earl Sorrells.

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
