## [Decision Letter · Decision Letter 0]

5 Aug 2022

PONE-D-22-20157QTL mapping of seedling and field resistance to stem rust in DAKIYE/Reichenbachii durum wheat populationPLOS ONE

Dear Dr. Megerssa,

Thank you for submitting your manuscript to PLOS ONE. After careful consideration, we feel that it has merit but does not fully meet PLOS ONE’s publication criteria as it currently stands. Therefore, we invite you to submit a revised version of the manuscript that addresses the points raised during the review process.

 The reviewers raised helpful comments/suggestions for improving the manuscript.  Please address them in the revised version.  

We look forward to receiving your revised manuscript.

Kind regards,

Dengcai Liu, PhD

Academic Editor

PLOS ONE

Journal Requirements:

"This study was supported by the DGGW project funded by the UK Aid from the British People and the Bill & Melinda Gates Foundation. The award number is OPP1133199."

"The authors wish to extend deepest gratitude to the DGGW project of Cornell University funded by the UK Aid from the British People and the Bill & Melinda Gates Foundation for funding this study. The authors are also thankful to the EIAR for the leave of absence and other supports to the student during the study. We are also grateful to collaborators from CIMMYT, EIAR, KALRO, and USDA-ARS, Eastern Regional Small Grains Genotyping Lab, Raleigh, NC. The authors also would like to extend sincere appreciation to Bekele Abiyo Bedada Girma and Ayele Badebo for processing the domestic quarantine after seed import."

"This study was supported by the DGGW project funded by the UK Aid from the British People and the Bill & Melinda Gates Foundation. The award number is OPP1133199."

Reviewers' comments:

Reviewer's Responses to Questions

**Comments to the Author**

1. Is the manuscript technically sound, and do the data support the conclusions?

Reviewer #1: Yes

Reviewer #2: Yes

2. Has the statistical analysis been performed appropriately and rigorously? 

Reviewer #1: Yes

Reviewer #2: Yes

3. Have the authors made all data underlying the findings in their manuscript fully available?

Reviewer #1: Yes

Reviewer #2: Yes

4. Is the manuscript presented in an intelligible fashion and written in standard English?

Reviewer #1: Yes

Reviewer #2: Yes

5. Review Comments to the Author

Reviewer #1: This manuscript by Megerssa et al. describes the QTLs of stem rust resistance in durum wheat genotypes using SNP genotyping. The results showed the resistant genotype Reichenbachii could carry three QTLs on different chromosomes. None of the QTL identified except the 3B QTL were consistent for the races evaluated and across environments. Overall, this paper is a useful resource for wheat stem rust research.

The comments are as follows:

1. If possible I suggest an additional cytological study is needed to clarify the potential chromosomal rearrangement in the present study.

2. Plenty of figs and tables are present in the main text, I suggest to move some of them less important to the supplementary online materials

3. Abstract: Introductory remark, the objectives of the study & methodology need elaboration.

4. L30 ETMS19 and KNMS19 represent the environments not races. This statement will confuse the authors.

5. In the methodology part, the reasons for using races JRCQC and TTRTF to conduct testing should be mentioned.

6. L89 The full name for GID?

7. L100 the conditions (e.g. temperature, humidity, light ) in the greenhouse should present.

8. L116-117, briefly describe the Stakman infection types (ITs), and the reasons to linearize the 0-4 scale to a 0 -9 scale

9. L124 what a plot stand for, it is not clear

10. L199-200 it is not clear?

11. Line 281 30MSMR?

12. Some recent publications should be cited in the MS; the format of reference needs improvement.

Reviewer #2: This manuscript by Megerssa et al. describes QTL mapping of seedling and field resistance to stem rust in DAKIYE/Reichenbachii durum wheat population. They identify three QTL on chromosomes 3B, 4B and 7B that were contributed by the resistant parent Reichenbachii. The results are of interest and relevance to the readership of PLoS One, the experiments are sound and the manuscript is generally well written.

There are several concerns which need to be addressed before the manuscript can be published.

1. I concerned about the quantity / density of available markers used in the current study. The marker density used here is relatively low. It is strange for me that there are many blank regions without markers on many chromosomes (figure 6), and no markers on chromosome 7A.

2. The positions of QTL QSr.cnl-3B mentioned in line332, line354, and Line434 are the peak positions of QSr.cnl-3B, but not the interval (between two borders). The real interval of QTL QSr.cnl-3B could be larger, but not 64-67 cM mentioned in line 434.

3. “Data not shown” appear several times in the text. It would be better to provide the data, such as the data about “the QTL and environment interaction” (line 377). Or delete a description like this.

4. The figures can be rearranged (too many figures): combined some of them or move to supplementary. In figure 9, chromosomes 1B , 6A was not mentioned in the text, delete?

6. PLOS authors have the option to publish the peer review history of their article (what does this mean?). If published, this will include your full peer review and any attached files.

Reviewer #1: No

Reviewer #2: No

---

## [Author Response · Author response to Decision Letter 0]

17 Aug 2022

Responses for the academic editor

Dear editor, thank you for your time, for the comments and suggestions on our manuscript. Your comments and suggestions helped us improve our manuscript. Below are the responses for some of the concerns and comments. 

Comment 1. Please ensure that your manuscript meets PLOS ONE's style requirements, including those for file naming. The PLOS ONE style templates can be found at 

Response comment 1: The manuscript I submitted on the first round does not fully meet the style of PLOS ONE. I looked at the PLOS ONE website. The formatting that I saw looks different from the link I received on your past email. Thank you for sending this link I corrected the style and file naming based on PLOS ONE style.

The affiliation of the authors was not based on PLOS ONE style, because I missed this part on the website. Thanks for the comment, I corrected it based on the journal style.

Comment 2: Please state what role the funders took in the study. If the funders had no role, please state: "The funders had no role in study design, data collection and analysis, decision to publish, or preparation of the manuscript." If this statement is not correct you must amend it as needed. Please include this amended Role of Funder statement in your cover letter; we will change the online submission form on your behalf.

Response comment 2: Thanks for the comment I will state the role of the funders in my cover letter. The funders provided me with the fund to pursue my study. However, they have no role in the design of the study, data collection, analysis, the funders have no publication decision except that the paper must be published in an open access journal.

Comment 3: We note that you have provided funding information that is not currently declared in your Funding Statement. However, funding information should not appear in the Acknowledgments section or other areas of your manuscript. We will only publish funding information present in the Funding Statement section of the online submission form. 

Please remove any funding-related text from the manuscript and let us know how you would like to update your Funding Statement.

Response comment 3: I provided the funding statement during my submission but, the project was phased out after I finalized my study. There may be a chance to get publication fee from this project. I contacted one of my advisors, the project representative, if I can still be able to get funding for publication and learnt that there is a possibility.

Thank you for the comment, I removed the funding statement from the acknowledgement.

Comment 3.1. Please include your amended statements within your cover letter; we will change the online submission form on your behalf.

"This study was supported by the DGGW project funded by the UK Aid from the British People and the Bill & Melinda Gates Foundation. The award number was OPP1133199."

Response Comment 3.1. Thank you, editor, this funding statement should be there because my study was supported by the UK Aid from the British People and the Bill & Melinda Gates Foundation. The award number was OPP1133199. The only thing different at present is, this project is phased out. I am not sure if I am still able to get publication fee from this project. However, I contacted my advisor about that and she will let me know after she communicated the Gates foundation. If I am unable to pay the publication fee from this project, we can send the invoice to her and she will handle it from a related project.

Comment 4: We note that you have included the phrase “data not shown” in your manuscript. Unfortunately, this does not meet our data sharing requirements. PLOS does not permit references to inaccessible data. We require that authors provide all relevant data within the paper, Supporting Information files, or in an acceptable, public repository. Please add a citation to support this phrase or upload the data that corresponds with these findings to a stable repository (such as Figshare or Dryad) and provide and URLs, DOIs, or accession numbers that may be used to access these data. Or, if the data are not a core part of the research being presented in your study, we ask that you remove the phrase that refers to these data.

Response Comment 4: Dear editor, thank you for the comment about the phrase “data not shown”. As per your suggestion, I included the analysis result as supplementary table that replaces this phrase and removed the other one which is not relevant to be included. 

For line 303: the “data not shown” phrase should be deleted because lines and or markers were removed from the data sets based on the different filtering criteria applied. 

For line 374, 395, 449, 503: supplementary table was prepared for the QTL effect and QTL by Environment interaction. Data not shown on line 431 deleted because it is on the filtering step and of no use to include.

 

Responses for Reviewer 1

Dear Reviewer, thank you for your time and comment. We valued your comment to improve our manuscript. And the responses for some of the concerns raised are mentioned below and comments are included in the manuscript.

Comment 1. If possible, I suggest an additional cytological study is needed to clarify the potential chromosomal rearrangement in the present study.

Response 1: Dear reviewer, thank you for the comment. We discussed about this point with my major advisor and as my study period was ended, he suggested me to find a cytologist in my home country Ethiopia and couldn’t find one and I heard that the only cytologist in the University at the Capital city has recently passed away. Because of that it became impossible to do it here for now. However, I am planning to discuss with other partners who have the capacity to do the cytological study together.

Comment 2. Plenty of figs and tables are present in the main text, I suggest to move some of them less important to the supplementary online materials.

Response 2. Dear Reviewer, thanks for the comment. I moved some to supplementary materials and deleted two figures which may be less important.

Comment 3. Abstract: Introductory remark, the objectives of the study & methodology need elaboration.

Response 3. Thanks for the comment on the abstract. I tried to make the introductory remark, objectives and methodology a bit clear although there is a limitation on the numbers of words to use on abstract. 

Comment 4. L30 ETMS19 and KNMS19 represent the environments not races. This statement will confuse the authors.

Response 4: Thank you for the comment the statement in this line (currently on L44) is corrected to avoid confusion.

Comment 5. In the methodology part, the reasons for using races JRCQC and TTRTF to conduct testing should be mentioned.

Response 5: Thank you for the comment, I included the reason for using these two races in the methodology (L107 to L109). At first, I thought the one that I explained in the introduction may explain that (L74 to L76). 

 

Comment 6. L89 The full name for GID?

Response 6: Thanks for the comment, the name is included in L103 (Genotype Identification, GID).

Comment 7. L100 the conditions (e.g. temperature, humidity, light) in the greenhouse should present.

Response 7: Thank you for the comment this information is included (L118 to L120).

Comment 8. L116-117, briefly describe the Stakman infection types (ITs), and the reasons to linearize the 0-4 scale to a 0 -9 scale. 

Response 8: Thank you for the comment, the Stakman infection types and the reasons for conversion of 0 - 4 to 0-9 scale was explained on L135 to L140.

Comment 9. L124 what a plot stand for, it is not clear

Response 9: Thanks for the comment, it is true the plot size was not mentioned as clear. I corrected it and also arranged the order of sentences to make it a bit understandable.

Comment 10. L199-200 it is not clear?

Response 9: Thank you for the comment, the genotyping and SNP calling was done in North Carolina and one of the co-authors who contributed this part of the methodology amended the statement as: 

L224 to L 225 : “ Single-ended 100bp read length Illumina (San Diego, CA) sequencing was performed on a Novaseq 6000 instrument” 

Comment 11. Line 281 30MSMR?

Response 11: Thank you for the comment, this is an error. I meant 30MRMS and corrected on the manuscript (L306). 

Comment 12. Some recent publications should be cited in the MS; the format of reference needs improvement.

Response 12: Thank you for the comment, it is true, some of the references are old. But the old ones like the Stakman scoring, Roelfs, Bechere et al. are the only references present for the issues discussed in the MS.

 

Responses for Reviewer 2

Dear Reviewer thanks for your time and comment. We valued your comment to improve our manuscript. The responses for some of the concerns raised are mentioned below and comments are included in the manuscript.

There are several concerns which need to be addressed before the manuscript can be published.

1. I concerned about the quantity / density of available markers used in the current study. The marker density used here is relatively low. It is strange for me that there are many blank regions without markers on many chromosomes (figure 6), and no markers on chromosome 7A.

Response 1: Thanks for the comments. It is true, the marker density was low. It was initially 7418 markers after filtering out markers with 20% missing data and 10% heterozygous calls. During the analysis, we applied several filtering steps mentioned on L231 to 252 (on the revised manuscript) to get appropriate/reasonable genetic map. On this step we lost several markers. Some of the chromosomes like Chr. 7A had the lowest marker density from the start and they were lost during the filtering steps. The segregation distortion was high but the population was RIL (F9). Because of this we recommended cytological study as a follow up study and also the resistant parent was from a very old collection.

2. The positions of QTL QSr.cnl-3B mentioned in line332, line354, and Line434 are the peak positions of QSr.cnl-3B, but not the interval (between two borders). The real interval of QTL QSr.cnl-3B could be larger, but not 64-67 cM mentioned in line 434.

Response 2: Thank you for the comment. It is true that is not the interval. The position of this QTL was 66 cM for KNMS19 and 67 cM for ETMS19 and I corrected that.

3. “Data not shown” appear several times in the text. It would be better to provide the data, such as the data about “the QTL and environment interaction” (line 377). Or delete a description like this.

Response 3: Thanks for the comment, similar comment was raised by Reviewer 1. I provided the result that shows the “QTL and environment interaction” as a supplementary table and removed the data not shown phrase.

4. The figures can be rearranged (too many figures): combined some of them or move to supplementary. In figure 9, chromosomes 1B, 6A was not mentioned in the text, delete?

Response 4: Thank you for the comment. I moved some of the Figures to supplementary Figures and deleted two figures that could be less important. I plotted the LOD score plot of these selected chromosomes to show the presence of below threshold peak signals.

---

## [Editor Report · Decision Letter 1]

22 Aug 2022

QTL mapping of seedling and field resistance to stem rust in DAKIYE/Reichenbachii durum wheat population

PONE-D-22-20157R1

Dear Dr.  Megerssa,

We’re pleased to inform you that your manuscript has been judged scientifically suitable for publication and will be formally accepted for publication once it meets all outstanding technical requirements.

Kind regards,

Dengcai Liu, PhD

Academic Editor

PLOS ONE
---

## [Editor Report · Acceptance letter]

2 Sep 2022

PONE-D-22-20157R1 

QTL mapping of seedling and field resistance to stem rust in DAKIYE/Reichenbachii durum wheat population 

Dear Dr. Megerssa:

I'm pleased to inform you that your manuscript has been deemed suitable for publication in PLOS ONE. Congratulations! Your manuscript is now with our production department. 

Kind regards, 

on behalf of

Dr. Dengcai Liu 

Academic Editor

PLOS ONE